# What factors contribute to the Scapular Assistance Test result? A classification and regression tree approach

**Larissa Pechincha Ribeiro**[1], **Rodrigo Py Gonçalves Barreto**[1], **Ricardo Augusto Souza Fernandes**[2], **Paula Rezende Camargo**[1] *

**1** Laboratory of Analysis and Intervention of the Shoulder Complex, Department of Physical Therapy, Universidade Federal de São Carlos, São Carlos, Brazil, **2** Department of Electrical Engineering, Universidade Federal de São Carlos, São Carlos, Brazil

* prcamargo@ufscar.br

**Data Availability Statement:** All relevant data are within the paper and its Supporting Information files.

## Abstract

The aim of this study was to determine predictive factors related to the Scapular Assistance Test in individuals with shoulder pain during arm elevation, and to analyze how these predictors interact in a nonlinear manner to discriminate the result of a positive and negative Scapular Assistance Test. Eighty-four individuals with shoulder pain with positive (n = 47, average age 38.4 years) and negative (n = 37, average age 37.8 years) Scapular Assistance Test completed the study. Demographic data, affected shoulder, pain duration, pain at rest, angular onset of pain, scapular dyskinesis, serratus anterior and lower trapezius muscle strength, Disabilities of Arm, Shoulder and Hand questionnaire and Pain Catastrophizing Scale were assessed in all participants. The Classification and Regression Tree analysis was used to determine which factors would predict the occurrence of a positive or negative Scapular Assistance Test and possible interactions among them. The resulting tree presented seven levels that combine the following variables: angular onset of pain, presence of scapular dyskinesis, pain catastrophizing, serratus anterior and lower trapezius muscle strength. The angular onset of pain during arm elevation was the main predictor of a positive Scapular Assistance Test selected by the Classification and Regression Tree. This study indicates that the Scapular Assistance Test result may be explained not only by biomechanical variables, but also by psychological factors. Disability of the upper limbs does not seem to contribute to the Scapular Assistance Test result.

## Introduction

Although alterations in scapular motion can be observed in asymptomatic shoulders, these are more commonly described in the presence of painful conditions [1–4]. Because of the conflicting evidence regarding the causal relationship of scapular altered motion and pain, this issue is still under debate [5].

Symptom modification tests are frequently used to assess possible association between scapular motion and pain. This approach aligns well with recent proposals on applying a

**Funding:** This project was funded by Coordenação de Aperfeiçoamento de Pessoal de Nível Superior and Conselho Nacional de Desenvolvimento Científico e Tecnológico (144436/2019-1). The funders had no role in study design, data collection and analysis, decision to publish, or preparation of the manuscript.

**Competing interests:** The authors have declared that no competing interests exist.

classification system that is based on movement-impairment rather than pathoanatomical findings [6, 7]. The Scapular Assistance Test (SAT) is one of these tests in which the scapula is manually assisted by the examiner into upward rotation and posterior tilt while the patient actively elevates the arm [8, 9]. The SAT is considered positive when a reduction of at least two points on the 11-point numerical pain rating scale occurs during assisted elevation as compared to elevation without assistance [8]. A positive SAT may suggest inadequate strength and/or activation of the serratus anterior and lower trapezius [8].

A recent study [10] reported a large variability in the clinical presentation in patients with shoulder pain, and described that factors related to range of motion, rotator cuff function and scapular motion were associated with shoulder pain, dysfunction of the upper limbs, and psychological aspects. The clinical presentation of individuals with shoulder pain is usually characterized by painful elevation of the arm, dysfunction, altered scapular motion, non-optimal activation of the scapular muscles, and more recently, by psychological aspects such as pain catastrophizing [1, 2, 11–13]. The heterogeneity of the clinical presentation and interactions among the different factors have not been previously investigated and may influence the result of the SAT. This study will enhance the knowledge to better understand how the SAT may guide the clinical decision-making in the management of individuals with shoulder pain.

Therefore, the aim of this study was to determine predictive factors related to the SAT in individuals with shoulder pain during arm elevation, and to analyze how these predictors interact in a nonlinear manner to discriminate the result of a positive and negative SAT.

## Materials and methods

### Participants and study design

This is a cross-sectional study. Individuals were included if they had atraumatic self-reported unilateral shoulder pain during flexion of the arm for at least four weeks since first onset and active arm flexion (~150°) as measured by a digital inclinometer (Acumar$^{TM}$ Lafayette Instrument Company, Lafayette, IN). The participants were selected from a larger project that was conducted by the same research group. Recruitment was performed by advertisements on local websites and printed flyers at the university and in the community. Participants were excluded for any of the following reasons: history of fracture or previous surgery in the upper limbs, presence of neck-related pain determined by the Spurling's and cervical quadrant tests [14, 15], shoulder pain reproduced by the Upper Limb Tension Test for the median nerve [16], and recurrent glenohumeral joint dislocations in the last two years.

The main project in which this study is part of was submitted and approved by the Human Research Ethics Committee of Universidade Federal de São Carlos (protocol number 1.394.925). All participants received verbal and written explanation of the study, and those who agreed to participate signed an informed consent.

### Examination procedures

All participants were examined by one of two physical therapists with at least 3 years of clinical and research experience related to shoulder disorders, and with specific training in shoulder assessment and rehabilitation. After having the demographics information measured and recorded, SAT, pain intensity at rest, angular onset of pain during arm elevation, scapular dyskinesis, strength of the serratus anterior and lower trapezius, Disabilities of Arm, Shoulder and Hand (DASH) questionnaire and Pain Catastrophizing Scale (PCS) were assessed in all participants. All outcomes were collected at Laboratory of Analysis and Intervention of the Shoulder Complex, Department of Physical Therapy in the same day by the same examiner.

## Scapular Assistance Test

Initially, the individual was asked to elevate the arm in the sagittal plane and rate the shoulder pain on the 11-point numerical pain rating scale. Next, the patient was asked to again elevate the arm and rate the pain while the examiner assisted the scapular upward rotation by pushing upward and laterally on the inferior angle, and the scapular posterior tilt by pulling the superior aspect of the scapula [8, 17]. The maneuver was performed once. The test was considered positive when individuals reported a decrease in shoulder pain of two or more points on the 11-point numerical pain rating scale during the assisted elevation as compared to the elevation without assistance [8].

## Pain at rest and angular onset of pain

Presence of pain was assessed by asking the individuals if shoulder pain was present (yes/no) at rest at the moment of data collection. The angular onset of pain during elevation of the arm was measured with a digital inclinometer (Acumar[TM] Lafayette Instrument Company, Lafayette, IN), which has been shown to be reliable [18]. Individuals were instructed to assume a relaxed standing position with the arms at their side, and then raise the arm in the sagittal plane until the onset of shoulder pain [11]. The inclinometer was placed distally on the humeral shaft for registration of the angle. Only one trial was performed.

## Scapular dyskinesis

Scapular dyskinesis was assessed by clinical observation of scapular motion during active, bilateral, and non-weighted elevation of the arm in the sagittal plane. It was considered present (yes) when the prominence of the medial scapular border, inferior angle or rapid scapular downward rotation was observed in 3/5 trials of arm elevation in the sagittal plane [19]. Scapular dyskinesis was considered absent (no) when there were no abnormalities in scapular motion during arm elevation [19]. Classifying scapular dyskinesis as present or absent provides better inter-rater percent agreement (79%) [20].

## Muscle strength

Serratus anterior (SA) and lower trapezius (LT) muscle strength was measured using a hand-held dynamometer (Lafayette Instrument Company, Lafayette, IN, USA). These muscles were tested for being the main contributors for scapular upward rotation and posterior tilt [21, 22], which are the assisted motions during execution of the SAT. To assess the SA, the participants were in supine with the elbow and arm at 90° [23–25]. The dynamometer was placed on the elbow and the force was applied to the ulna perpendicular to the table. Individuals were instructed to protract their shoulder [24]. To assess the LT, individuals were in prone with the elbow in extension and arm at 140° of abduction. The dynamometer was placed on the lateral third of the scapula between the acromion and the root of the spine. Individuals were instructed to move their scapula in direction to the opposite hip while the examiner applied force in the superior and lateral directions parallel to the long axis of the humerus [24, 25].

Three submaximal repetitions of each test were performed for familiarization. Next, three 5-second repetitions of maximal isometric contractions with a 30 second-rest interval between repetitions were performed. Both tests' positions followed the recommendations of previous studies [23–25]. A standardized verbal encouragement to develop maximal strength in all contractions was given by the principal investigator in a consistent manner to all participants during the testing procedure. Resistance was manually applied by the examiner who had to keep a

constant resistance during the test. The order of assessment of each muscle was randomized. The average of the 3 repetitions was used for data analysis.

### Disability of the upper limbs

The Brazilian version of DASH (Disabilities of the Arm, Shoulder, and Hand) questionnaire was used to assess the disability of the upper limbs. It contains 30 questions that are scored on a 5-point rating scale. Final score ranges from 0 to 100. Higher scores indicate the worst possible condition [26]. The Brazilian version of the DASH has been shown to be a reliable instrument [26].

### Pain catastrophizing

Pain catastrophizing was measured with the Brazilian version of Pain Catastrophizing Scale (PCS). This score comprises 13 questions divided into 3 domains: magnification, helplessness, and rumination. The total score ranges from 0 to 52. Higher scores indicate more catastrophic thoughts [27]. Scores equal or higher than 30 points are considered high degree of catastrophizing [28]. PCS has adequate inter and intra-rater [27] reliability and has been used in patients with shoulder pain [13, 29].

### Statistical analysis

Statistical analysis was conducted with Statistical Package for the Social Sciences (SPSS Inc., Chicago, IL) version 25.0. The Kolmogorov-Smirnov test was used to check data distribution. Chi-squared tests were used to compare sex, affected shoulder, pain at rest and scapular dyskinesis. Unpaired Student's t-tests were used to compare height, body mass, serratus anterior muscle strength, DASH and PCS between individuals with positive and negative SAT. Mann-Whitney tests were used to compare age, duration of pain, angular onset of pain and lower trapezius muscle strength between individuals with positive and negative SAT.

Descriptive analyses were used to characterize the sample in relation to the outcome variable of interest the SAT and its selected potential predictors (pain at rest, angular onset of pain, serratus anterior strength, lower trapezius strength, scapular dyskinesis, DASH and PCS). Predictors and characteristics of the participants were compared between those with a positive and negative SAT. The Classification and Regression Tree (CART) analysis was used to determine which factors would predict the occurrence of a positive or negative SAT and possible interactions among them. A CART is a multivariate, nonparametric classification (regression) model that develops a decision tree by successive divisions of the initial set of data, until further divisions are not possible or until pre-established criteria for tree growth are reached [30]. For each of these divisions, all possible predictors and their respective cutoff points are considered to establish the predictor that best classifies the individuals into each of the outcome categories [30, 31]. The order of entry of predictors in the model illustrates hierarchically the strength of association between each predictor and outcome variable, and subsequent divisions identify possible interaction among predictors. The choice of the CART to analyze the data was based on its robust analysis, which captures nonlinear relationships between predictors and produces results easily applied in clinical practice [31, 32].

For this study, the Weka open-source software was used. The CART was parameterized to present a confidence factor not less than 0.95. The leave-one-out cross-validation process was considered to overcome the possibility of overfitting during the learning [33–35]. More specifically, in this cross-validation process, one sample of the set of data is used to validate the CART, while the remaining data are used for training.

To facilitate clinical comprehension of the prediction model produced by the CART, the dependent variable (SAT) was dichotomized as percentiles corresponding to a positive and negative SAT of the sample's distribution. The area under the receiver-operating characteristic (ROC) curve was calculated to verify the performance of the prediction model. A significance level of 0.05 was established to verify whether the area under the ROC curve was different from 0.5, which would indicate that the model was accurate to predict the outcome categories.

## Results

A total of 236 individuals were recruited. Fig 1 brings the flow chart of the study. Eighty-four individuals completed the study. The characteristics of the individuals and predictors are presented in Table 1. Individuals with a positive SAT presented angular onset of pain higher in the range of arm flexion as compared to those with a negative SAT (p<0.05, mean difference: 20.35˚, 95% CI: 4.70˚; 36.01˚). Individuals with a positive SAT presented less disability of the upper limbs when compared with individuals with negative SAT (p<0.05, mean difference: 7.58 points, 95% CI: 0.08; 15.08). Individuals with a positive SAT also presented lower score in the PCS than individuals with a negative SAT (p<0.05, mean difference: 7.62 points, 95% CI: 2.56; 12.68). There was no difference for the other outcomes (Table 1).

The classification tree selected the angular onset of pain during arm elevation as the first predictor variable of the SAT (Fig 2). Individuals with an angular onset of pain below or equal 65˚ of arm elevation will present a negative SAT. However, this variable alone did not entirely explain the occurrence of a positive SAT, and scapular dyskinesis was selected as the second predictor. Details about tree divisions, with the respective predictors' cutoff points and the number and percentage of individuals classified in each subgroup according to the selected predictors are presented in Fig 2.

The predictive model correctly classified 47 of the 47 individuals with positive SAT, and 31 of the 37 individuals with negative SAT. Overall, the predictive model obtained an average accuracy of 92.9% and ROC area of 0.96.

## Discussion

The findings of this study demonstrated that a positive SAT may be influenced by different interactions of contributing factors. The angular onset of pain during arm elevation was the main predictor of a positive SAT selected by the CART. However, this variable alone did not entirely explain the occurrence of a positive SAT, and other predictors (scapular dyskinesis, pain catastrophizing, pain at rest and strength of the serratus anterior and lower trapezius muscles) were selected by the classification tree. Interestingly, pain catastrophizing was selected multiple times and may be considered an important predictor for a positive SAT. The cutoff points defined by the CART analysis may help clinicians to target factors that may be related to the occurrence of a positive SAT.

Those individuals with an angular onset of pain below 65˚ of arm elevation had a negative SAT. This finding is not surprising since contribution of the scapula is low at the initial ranges of arm elevation [36, 37]. In addition, pain and functionality of the arm may be better perceived above 65˚ where most daily and labor activities are performed [38], and contribute to a positive SAT from the interaction with the other predictors used in this study. The presence of dyskinesis combined with low pain catastrophizing (≤ 6 points at the PCS) contributed to a positive SAT above 65˚. This indicates that the pain of these patients is likely influenced by altered scapular motion [9]. However, due to the chronicity of pain of the individuals in the current study, central impairments on pain modulation may be present and different phenotypes of people with shoulder pain may exist [39]. Taking all together, it is valid to observe

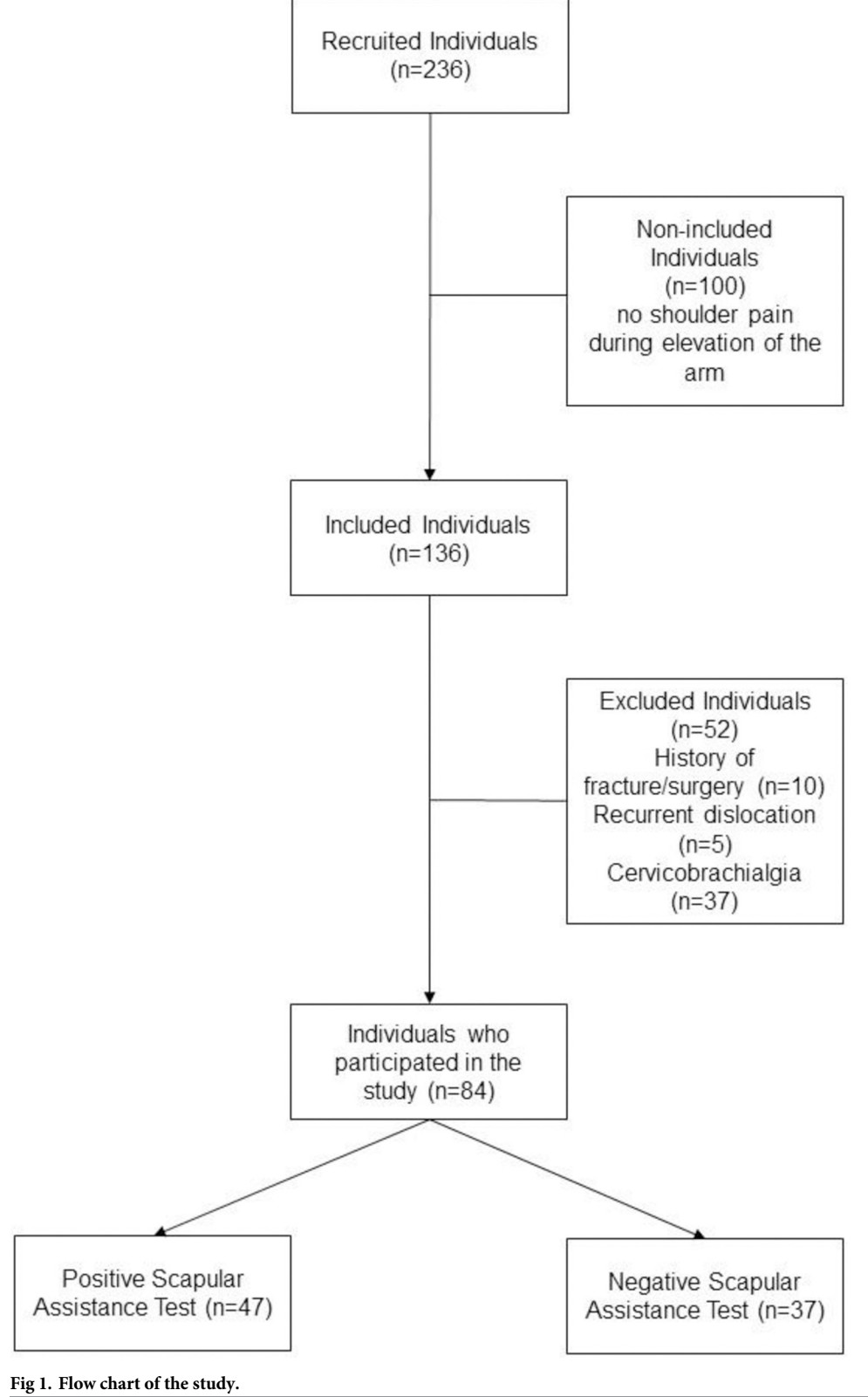

**Fig 1. Flow chart of the study.**

**Table 1. Characteristics of the participants and predictive factors.**

| | Positive SAT group (n = 47) | Negative SAT group (n = 37) | Mean Difference (95% Confidence Interval) | P-value |
|---|---|---|---|---|
| Sex[a] | 17 women | 21 women | - | 0.060 |
| | 30 men | 16 men | | |
| Age, years[b] | 38.49 ±14.0 | 37.84 ± 12.1 | -0.65 (-6.43, 5.13) | 0.910 |
| Body mass, kg[c] | 73.5 ± 10.22 | 72.5 ± 14.20 | 2.66 (-6.28, 4.32) | 0.724 |
| Height, m[c] | 1.72 ± 0.08 | 1.71 ± 0.10 | -0.01 (-0.05, 0.03) | 0.613 |
| Affected shoulder[a] | 32 dominants | 27 dominants | - | 0.627 |
| | 15 non-dominants | 10 non-dominants | | |
| Duration of pain, months[b] | 28.51 ± 48.0 | 26.6 ± 38.5 | -1.86 (-21.14, 17.41) | 0.765 |
| Pain at rest, n (%)[a] | 14 (29.8%) yes | 13 (35.1%) yes | - | 0.602 |
| | 33 (70.2%) no | 24 (64.9%) no | | |
| Angular onset of pain, degrees[b] | 134.09 (30.9) | 113.73 (41.2) | -20.35 (-36.01, -4.70) | 0.019* |
| Scapular dyskinesis, n (%)[a] | 35 (74.5%) yes | 24 (64.9%) yes | - | 0.339 |
| | 12 (25.5%) no | 13 (35.1%) no | | |
| Serratus Anterior, Kgf[c] | 17.82 ± 6.23 | 16.50 ± 6.47 | -1.33 (-4.09, 1.41) | 0.337 |
| Lower Trapezius, Kgf[b] | 9.83 ± 4.03 | 8.1 ± 4.46 | -1.66 (-3.52, 0.19) | 0.080 |
| DASH (0–100)[c] | 23.57 ± 15.25 | 31.16 ± 19.2 | 7.58 (0.08, 15.0) | 0.047* |
| PCS (0–52)[c] | 16.51 ± 11.0 | 24.14 ± 12.2 | 7.62 (2.49, 12.75) | 0.004* |

Results are mean ± standard deviation.

*P<0.05, when statistically significant. Abbreviations: DASH, Disabilities of the Arm, Shoulder, and Hand; PCS, Pain Catastrophizing Scale. Higher scores in the DASH and PCS indicate worse disability and greater catastrophization, respectively. Kgf, kilogram force.

[a]: chi-squared tests.

[b]: Mann-Whitney tests.

[c]: Student's t tests.

other factors besides the biomechanical ones. Individuals with score above 6 points in the PCS, presence of pain at rest, and strength of the serratus anterior below 11.3 kilograms force also had higher likelihood of presenting a positive SAT.

Pain catastrophizing levels above 30 points indicate individuals susceptible to be catastrophizers [28]. Our study shows that mild levels of pain catastrophizing may moderate the probability of a positive SAT. For example, increased pain catastrophizing levels combined with intermediate values of serratus anterior strength (11.3 to 17.8 kilograms force) were associated with a negative SAT, but lower pain catastrophizing levels combined with higher values of serratus anterior strength (> 17.8 kilograms force) were associated with a positive SAT. The serratus anterior is an important scapular mover [21], and any change in pain or range of shoulder elevation during the assistance of the scapula has already been associated as a predictor of better outcome after physical therapy management for shoulder pain [40]. For individuals with pain during movement of the arm and scapular dyskinesis, pain catastrophizing may be more relevant as higher levels of pain catastrophizing were associated with a negative SAT.

When scapular dyskinesis is not present, strength of the lower trapezius should be assessed along with pain catastrophizing and strength of the serratus anterior. It is worthy to note that pain at rest was only selected when scapular dyskinesis was present, and that strength of the lower trapezius was only selected in the absence of scapular dyskinesis. Based on the muscle position, the lower trapezius muscle is most aligned to execute scapular upward rotation during arm abduction when compared to flexion [22]. However, SAT was performed during flexion of the arm in the current study. Future research may apply the SAT during abduction to investigate possible deficits in the lower trapezius muscle strength. Kibler [9] suggested that a

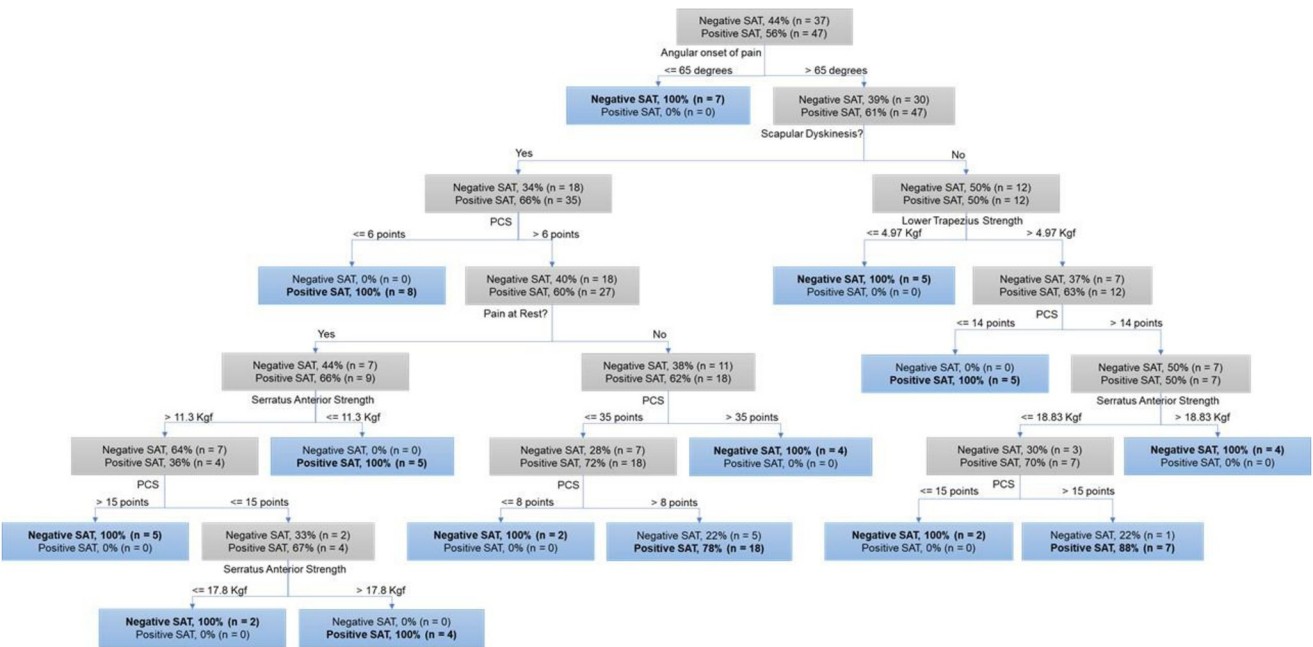

**Fig 2. Classification tree of the occurrence of a positive and negative Scapular Assistance Test.** The bolded category in each node corresponds to the predicted category. Abbreviations: SAT, Scapular Assistance Test; DASH, Disabilities of the Arm, Shoulder and Hand; PCS, Pain Catastrophizing Scale. Blue boxes indicate leaves of the tree. Gray boxes indicate intermediate nodes of the tree.

positive SAT was related to a deficit in muscle activation or strength of the scapular muscles. However, recent studies [25, 41] did not identify differences in the serratus anterior and lower trapezius muscle strength between those with a positive and negative SAT. In the current study, muscle strength of the serratus anterior and lower trapezius alone could not explain the occurrence of a positive SAT. The inclusion of pain catastrophizing by the CART suggests that the SAT may be influenced not only by biomechanical variables but also by psychological factors.

Although the participants in this study presented high muscle strength, we cannot talk about muscle activation. Serratus anterior and lower trapezius muscles may not be properly activated during elevation of the arm. Clinicians should also be aware that previous studies have shown that individuals with a positive SAT are more likely to present greater scapular anterior tilt (Ribeiro et al., 2020) and pectoralis minor tightness (Grimes et al., 2020).

Negative psychological factors have been reported in individuals with shoulder pain. These factors include emotional aspects such as anxiety and depression, [42] kinesiophobia, [43] fear and avoidance beliefs [44] and pain catastrophization [29]. Recently, individuals with worse psychological profile were shown to present worse function and higher shoulder pain intensity [13]. In the present study, individuals with negative SAT presented higher pain catastrophizing thoughts when compared to individuals with a positive SAT. Clinicians should pay attention to individuals with a negative SAT as they may benefit of an approach that targets psychological factors in addition to therapeutic exercises. Further research should investigate the influence of other psychological factors, such as kinesiophobia, self-efficacy, optimism and resilience that may also predict the SAT result.

From the resulting tree, the cutoff points adjusted during the training stage are accurate since the classification leaves (in blue) mostly separate the two classes accordingly. Analyzing the tree, two leaves presented a misclassification of 6 individuals: (i) 5 individuals in the leaf

that used the PCS with a cutoff point above 8 points; and (ii) 1 individual in the leaf that used the PCS with a cutoff point above 15 points. Both leaves are predominantly composed of individuals with positive SAT meaning that these leaves are better suited to classify individuals with positive SAT.

In summary, the resulting tree has 7 levels that combine the following variables: angular onset of pain, presence of scapular dyskinesis, pain catastrophizing, serratus anterior and lower trapezius muscle strength. Based on the results, these variables may be considered the most relevant to distinguish individuals with a positive and negative SAT. The high accuracy of the model indicates that the classification of individuals in positive or negative SAT was not random, nor was it affected by the imbalance between the number of individuals with positive or negative SAT. It is important to note that patients with negative SAT presented worse function, but also higher levels of pain catastrophizing when compared to positive SAT. Function of the upper limbs as evaluated by the DASH was not a relevant variable in the tree. This study suggests that biomechanics may not explain the SAT result alone, but there may be influence of patients' belief in relation to their pain with regards to magnification, rumination and hopelessness. These findings are in line with another study [12] reporting that PCS explains function variance more than biological factors such as MRI data.

This study has some limitations. Although a 2-point decrease in pain intensity was considered for a positive SAT, the points on the numerical pain rating scale was not registered. The SAT was performed during flexion of the arm and our findings cannot be extrapolated to other planes of arm elevation. Some patients reported pain during execution of the muscle strength tests, and this may have influenced the results. Future studies should explore the contribution of other psychological factors to the contribution to SAT results such as positivism, resilience and self-efficacy. Finally, one should notice that this study does not allow a causal relationship to be made between the clinical variables.

## Conclusion

The SAT result was influenced by different interactions of contributing factors, such as angular onset of pain during arm elevation, scapular dyskinesis, pain catastrophizing, pain at rest and serratus anterior and lower trapezius muscle strength. This indicates that the SAT result may be explained not only by biomechanical variables, but also by psychological factors. Disability of the upper limbs does not seem to contribute to the SAT result.

## Supporting information

**S1 Data.**
(XLSX)

## Author Contributions

**Conceptualization:** Larissa Pechincha Ribeiro, Rodrigo Py Gonçalves Barreto, Paula Rezende Camargo.

**Data curation:** Larissa Pechincha Ribeiro, Rodrigo Py Gonçalves Barreto, Ricardo Augusto Souza Fernandes.

**Formal analysis:** Ricardo Augusto Souza Fernandes.

**Investigation:** Larissa Pechincha Ribeiro, Rodrigo Py Gonçalves Barreto, Paula Rezende Camargo.

**Methodology:** Larissa Pechincha Ribeiro, Ricardo Augusto Souza Fernandes, Paula Rezende Camargo.

**Project administration:** Larissa Pechincha Ribeiro, Paula Rezende Camargo.

**Software:** Ricardo Augusto Souza Fernandes.

**Supervision:** Paula Rezende Camargo.

**Writing – original draft:** Larissa Pechincha Ribeiro, Ricardo Augusto Souza Fernandes, Paula Rezende Camargo.

**Writing – review & editing:** Larissa Pechincha Ribeiro, Rodrigo Py Gonçalves Barreto, Ricardo Augusto Souza Fernandes, Paula Rezende Camargo.

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
