## [Decision Letter · Decision Letter 0]

24 Jun 2022

PONE-D-21-38851

What factors contribute to the Scapular Assistance Test result? A classification and regression tree approach

PLOS ONE

Dear Dr. Pechincha Ribeiro,

Thank you for submitting your manuscript to PLOS ONE. After careful consideration, we feel that it has merit but does not fully meet PLOS ONE’s publication criteria as it currently stands. Therefore, we invite you to submit a revised version of the manuscript that addresses the points raised during the review process.

Please see the reports from four reviewers below. All reviewers appear to be positive about the contribution of the study, and have made specific suggestions for enhancing the reporting and clarity of the manuscript.

We look forward to receiving your revised manuscript.

Kind regards,

Hanna Landenmark

Staff Editor

PLOS ONE

“Funding: Coordenação de Aperfeiçoamento de Pessoal de Nível Superior and Conselho Nacional de Desenvolvimento Científico e Tecnológico (144436/2019-1).”

“We would like to thank Coordenação de Aperfeiçoamento de Pessoal de Nível Superior and Conselho Nacional de Desenvolvimento Científico e Tecnológico (144436/2019-1).”

“Funding: Coordenação de Aperfeiçoamento de Pessoal de Nível Superior and Conselho Nacional de Desenvolvimento Científico e Tecnológico (144436/2019-1).”

Reviewers' comments:

Reviewer's Responses to Questions

**Comments to the Author**

1. Is the manuscript technically sound, and do the data support the conclusions?

Reviewer #1: Partly

Reviewer #2: Partly

Reviewer #3: Partly

Reviewer #4: Yes

2. Has the statistical analysis been performed appropriately and rigorously? 

Reviewer #1: Yes

Reviewer #2: Yes

Reviewer #3: Yes

Reviewer #4: Yes

3. Have the authors made all data underlying the findings in their manuscript fully available?

Reviewer #1: Yes

Reviewer #2: Yes

Reviewer #3: Yes

Reviewer #4: No

4. Is the manuscript presented in an intelligible fashion and written in standard English?

Reviewer #1: Yes

Reviewer #2: Yes

Reviewer #3: Yes

Reviewer #4: Yes

5. Review Comments to the Author

Reviewer #1: Thank you for giving me the opportunity to review this manuscript. I think this research is fascinating, given that it presents a method for determining factors and their weightings in the outcome of a clinical test that is widely used in the clinical setting. Along with congratulating the authors for their work, I believe that there are sensitive points of the research that should be considered, about the relevance of the research and methodological aspects that will improve the robustness of the analyses discussed.

Introduction

-Please provide further justification of the other disorders described in lines 58 to 61 and how these should be considered in the context of SAT. One of the hypotheses that is inferred from the study is that these are predictive factors for the positive finding, so this contribution should be better substantiated. I suggest reviewing Nordqvist et al (2021) Physiotherapy.

-Please provide further justification for the use of SAT in your research. In recent times there has been some discussion about the assessment of scapular motion in health conditions in the shoulder complex. On the other hand, SAT outcomes are also in question (see Lange et al (2017), Physical Therapy in Sport).

Materials and methods

-Please provide further justification as to why the following were considered as inclusion criteria: if they had atraumatic self-reported unilateral shoulder pain during flexion of the arm for at least four weeks since first onset and full active arm flexion (~150°) as measured 75 by a clinical inclinometer.

-Please provide further details of the clinical inclinometer used.

-Please include references to the tests performed on lines 79 to 80.

-Please include more information on expert credentials. I suggest taking as a reference the article by Wainwright et al (2010) Journal of Physical Therapy.

-Please clarify which test was used to determine the presence of scapular dyskinesis. According to the reference decreed by the authors, the criterion of McClure et al. is used. However, according to CART, it is classified as yes/no according to Uhl. I suggest reviewing Uhl et al (2009) Arthroscopy: The Journal of Arthroscopic and Related Surgery DOI: 10.1016/j.arthro.2009.06.007

-Please clarify whether SAT or modified SAT was used. According to the reference provided, modified SAT was used, however, in the manuscript it is presented as SAT.

-In view of what was explained in the introduction, the reliability of the test results is unknown. If you could present reliability values, it would be of great value to clear doubts about reliability of scapular dyskinesis tests. I suggest reviewing Guerrero-Henríquez (2021) Journal of Manual and Manipulative Therapies DOI: 10.1080/10669817.2021.1972653 and also Lange et al (2017), Physical Therapy in Sport.

-Regarding muscle strength testing, please provide further justification as to why these muscles are tested and not others associated with scapular dyskinesis, such as middle or upper trapezius.

- For muscle strength testing, in the case of SAT-positive participants might have evidenced pain in LT strength testing, in what way did they control the appearance of symptomatology?

-Regarding the muscle strength test, the authors state that three repetitions were performed. Please clarify whether the value used in statistical analysis was the average of the records or the maximum value.

-Please provide an a priori statistical power analysis in relation to the sample analyzed.

-Please indicate the procedure for selecting CART validation and training data.

Results

-Please, with a wide reading audience of the manuscript in mind, improve the presentation of Table 1: the units of the variables should be presented without parentheses (e.g.: Age, years) as it confuses with SD values in subsequent columns. Please remove the acronym SD from the columns, since it is explained in the table caption. Please indicate which p-values correspond to t-test and which to Chi2 test. Submit p-values with 4 significant values.

-The ideas presented in lines 225 to 227 should be included in the discussion section.

Discussion

-The authors propose their discussion by establishing causal relationships between the variables. Given the context of the research, a causal relationship between the clinical variables analyzed is not adequately justified. I suggest toning down a bit the relationships that CART allows to conclude. I suggest reviewing Nogueira et al (2022) Wires Data Mining and Knowledge Discovery DOI: 10.1002/widm.1449, who takes a rather broad look at the problematization of causality and the use of different algorithms to determine it.

-I suggest that the authors review the limitations of their research in accordance with the comments made.

Reviewer #2: Review

What factors contribute to the Scapular Assistance Test result? A classification and regression tree approach

ABSTRACT

1. Describe the group of factors used in the analysis such as biomechanics, psychological ... what else?

2. The results about the angular of onset which was the main predictor should be addressed in the abstract.

INTRODUCTION

1. The first sentence of second paragraph should be rewritten. The pectoralis minor length is not other factors but it should be considered as the underlying factors influencing the abnormal motion of scapular.

2. The underlying impairments reported to contribute to scapular movements and then could influence SAT should be listed. The factors might also should be introduce in categories as the authors addressed in the discussion.

Methods

1. 150 degrees should not be considered as full active elevation

2. Please describe the test procedure of Scapular Assistance Test clearly. Did you have the participants perform movement before assisant was provide. How many times the participants have to perform elevation? When and how you evaluate the pain? Etc

3. Did you also assess the pain intensity during SAT. Please report the change of pain intensity in both positive and negative SAT groups.

4. The Scapular dyskinesis test was tested in which plane?

5. For muscle strength test, Describe the direction of movement that used for producing isometric contraction of SA and LT.

Results

1. Table 1 should present mean±SD and number (%)

Discussion

1. What is kgf?

2. If the scapular dyskinesis is not present, how the assisting force and direction applied during the SAT?

3. So the subject with high level of pain catastrophizing tend to have negative or positive results of SAT?

4. Please discuss more about clinical implication of the findings. Which factors or outcomes should be considered in people with positive SAT?

Overall

Please recheck the grammar and format of inserted references.

Reviewer #3: Review Comments to the Author

Thank you for the opportunity to review the manuscript. The study aims to investigate determine predictive factors related to the SAT in individuals with shoulder pain during arm elevation, and to analyze how these predictors interact in a nonlinear manner to discriminate the result of a positive and negative SAT. These factors are explored with a decision tree analysis.

The manuscript is well written and described in adequate details for the most part. Please find below my comments for feedback.

Methods&Results

・This study`s include criteria was “Individuals were included if they had atraumatic self-reported unilateral shoulder pain during flexion of the arm for at least four weeks since first onset and full active arm flexion (~150°) as measured by a clinical inclinometer”. You should add an explanation as to why this was the inclusion criteria. (Example. Non-traumatic or traumatic, duration of onset: 4weeks, etc.)

・This study described the limitation section. “Although a 2-point decrease in pain intensity was considered for a positive SAT, a register before and after the SAT was not registered”. The NRS score at baseline is likely to affect the SAT result. Is the NRS score not included in the inclusion criteria for this study? If it is not included in the uptake criteria, I think it should be added to the limitation section.

・In Results, the predictive model obtained an average accuracy of 92.9％ and ROC area of 0.96. This model indicated good accuracy, but split was complexity. Therefore, it is difficult to interpret this model. How were splits selected in the tree (entropy or complexity penalty)? Also, the splits seem a little nuanced, given that the same variable is used more than once. Can the authors write a little about whether reported splits would be clinically relevant or significant?

Discussion

・I believe that SAT positivity is a test to indicate scapular dysfunction, but the results of this study included SAT-positive participants with high muscle strength. The author may write some thoughts on how SAT positivity should be interpreted and utilized in clinical practice, including muscle strength?

・Looking at the PCS results (e.g., SAT negative with PCS score ≤ 8 points and > 35 points), I do not think the PCS score determines SAT positives and negatives. Please check my comments about decision tree complexity. If the PCS score does affect the SAT results, can the author speculate on the mechanism why this might be the case? Since the PCS score is also lower than the cutoff, it is by no means clear that a high PCS score affects the SAT results. PCS score is also lower than the cutoff value, so we cannot say that the high PCS score affects.

Reviewer #4: Thank you to the authors for their submission. The paper addresses an important clinical and research topic and I am happy to review this paper again after the authors have addressed my concerns.

I suggest to include some sample characterization in the abstract (at least age).

In the first paragraph of introduction, I suggest to bring in a more detailed way what is the “problem in science” (relevance).

Authors should consider to add more information from references 10 and 11 about Other variables as serratus anterior and lower trapezius muscle strenght in the introduction. It should be clear in the introduction whats the importance (what Fields knows and remining gap?) of adding these variables to the model. Only the pectoralis minor lenght was mentioned and this variable has not been evaluated.

The authors mentioned that all data were presetend in the manuscript. However, in addition to summary statistics, the data points behind means and standard deviation also should be avaiable as a requirement from the Journal.

I would suggest to add more information in the results description.

Lines 243-245: I would suggest to make this sentence/association clearer and also add the reference.

Line 248 – “the pain are likely dye to altered scapular motion” – I suggest to rewrite this sentence considering the several factos involved in the chronicity of atraumatic shoulder pain (patients had an average of 27 months of pain duration).

Lines 249 - 252: please, consider to make clearer the findinds interpretation: increased pain catastrophising, increased ? pain at rest, decreased serratus anterior strenght.

Lines 263 – 266: The study design and the findings presented may not support this suggetion.

Line 267: “mediating effect”: I suggest a clearer definition of this term.

6. PLOS authors have the option to publish the peer review history of their article (what does this mean?). If published, this will include your full peer review and any attached files.

Reviewer #1: No

Reviewer #2: No

Reviewer #3: **Yes: **Hayato Shigetoh

Reviewer #4: **Yes: **Denise Martineli Rossi

---

## [Author Response · Author response to Decision Letter 0]

26 Sep 2022

Dear editor and reviewers,

We appreciate the opportunity to re-submit this manuscript. Thank you for your time and effort in reviewing this manuscript. We appreciate all your comments and recommendations that led to great improvements in this study. We have done our best efforts to address your concerns and hope that you will reconsider this study for publication. Below are the comments followed by the actions to address the concerns and all changes in the manuscript were highlighted in gray.

Response: The manuscript meets the journal’s style requirements and file naming.

“Funding: Coordenação de Aperfeiçoamento de Pessoal de Nível Superior and Conselho Nacional de Desenvolvimento Científico e Tecnológico (144436/2019-1).”

Response: The funders had no role in the study. This information was added in the cover letter. 

“We would like to thank Coordenação de Aperfeiçoamento de Pessoal de Nível Superior and Conselho Nacional de Desenvolvimento Científico e Tecnológico (144436/2019-1).”

“Funding: Coordenação de Aperfeiçoamento de Pessoal de Nível Superior and Conselho Nacional de Desenvolvimento Científico e Tecnológico (144436/2019-1).”

Response: Funding-related text was removed from the manuscript.

Response: The amended statement was included in the cover letter.

Response: There are no restrictions on sharing data publicity. This is stated in the cover letter.

Reviewers' comments:

Reviewer's Responses to Questions

Comments to the Author

1. Is the manuscript technically sound, and do the data support the conclusions?

Reviewer #1: Partly

Reviewer #2: Partly

Reviewer #3: Partly

Reviewer #4: Yes

 2. Has the statistical analysis been performed appropriately and rigorously?

 Reviewer #1: Yes

Reviewer #2: Yes

Reviewer #3: Yes

Reviewer #4: Yes

 3. Have the authors made all data underlying the findings in their manuscript fully available?

Reviewer #1: Yes

Reviewer #2: Yes

Reviewer #3: Yes

Reviewer #4: No

 4. Is the manuscript presented in an intelligible fashion and written in standard English?

Reviewer #1: Yes

Reviewer #2: Yes

Reviewer #3: Yes

Reviewer #4: Yes

 5. Review Comments to the Author

Reviewer #1: Thank you for giving me the opportunity to review this manuscript. I think this research is fascinating, given that it presents a method for determining factors and their weightings in the outcome of a clinical test that is widely used in the clinical setting. Along with congratulating the authors for their work, I believe that there are sensitive points of the research that should be considered, about the relevance of the research and methodological aspects that will improve the robustness of the analyses discussed.

Response: Thank you for your time and effort in reviewing this manuscript. We hope we were able to address your concerns.

Introduction

- Please provide further justification of the other disorders described in lines 58 to 61 and how these should be considered in the context of SAT. One of the hypotheses that is inferred from the study is that these are predictive factors for the positive finding, so this contribution should be better substantiated. I suggest reviewing Nordqvist et al (2021) Physiotherapy.

Response: The introduction was worked to explain the predictive factors for a positive SAT (pages 4-5, lines 69-80).

- Please provide further justification for the use of SAT in your research. In recent times there has been some discussion about the assessment of scapular motion in health conditions in the shoulder complex. On the other hand, SAT outcomes are also in question (see Lange et al (2017), Physical Therapy in Sport).

Response: The introduction was rewritten to address your concerns (pages 4-5, lines 58-80).

Materials and methods

- Please provide further justification as to why the following were considered as inclusion criteria: if they had atraumatic self-reported unilateral shoulder pain during flexion of the arm for at least four weeks since first onset and full active arm flexion (~150°) as measured 75 by a clinical inclinometer.

Response: This study was part of a larger project, in which individuals with unilateral shoulder symptoms had the 3D scapular motion measured. Full active arm flexion (~150°) was needed for a complete set of kinematic data. This information was added on page 5 (lines 92-93).

- Please provide further details of the clinical inclinometer used.

Response: Details of the inclinometer are now provided on page 5 (line 91) and page 7 (lines 133-134).

- Please include references to the tests performed on lines 79 to 80.

Response: The references were added on page 5 (lines 97-98).

- Please include more information on expert credentials. I suggest taking as a reference the article by Wainwright et al (2010) Journal of Physical Therapy.

Response: More information was added on page 6 (lines 108-109).

- Please clarify which test was used to determine the presence of scapular dyskinesis. According to the reference decreed by the authors, the criterion of McClure et al. is used. However, according to CART, it is classified as yes/no according to Uhl. I suggest reviewing Uhl et al (2009) Arthroscopy: The Journal of Arthroscopic and Related Surgery DOI: 10.1016/j.arthro.2009.06.007

Response: Scapular dyskinesis was considered present (yes) when the prominence of the medial scapular border, inferior angle or rapid scapular downward rotation was observed in 3/5 trials of arm elevation, and absent (no) when there were no abnormalities in scapular motion during arm elevation. The sentence was rewritten to clarify how we defined scapular dyskinesis on page 7 (lines 142-147). The definition of dyskinesis as present or absent allows higher inter-rater percent agreement (79%) (Uhl et al., 2009).

- Please clarify whether SAT or modified SAT was used. According to the reference provided, modified SAT was used, however, in the manuscript it is presented as SAT.

Response: In fact, we used the modified SAT that considers scapular upward rotation and posterior tilt during execution of the test. However, we preferred to call it as SAT as this procedure is usually described in literature (Rabin et al., 2006; Rabin et al., 2018; Grimes et al., 2020).

- In view of what was explained in the introduction, the reliability of the test results is unknown. If you could present reliability values, it would be of great value to clear doubts about reliability of scapular dyskinesis tests. I suggest reviewing Guerrero-Henríquez (2021) Journal of Manual and Manipulative Therapies DOI: 10.1080/10669817.2021.1972653 and also Lange et al (2017), Physical Therapy in Sport.

Response: The two studies suggested by the reviewer indicate that classifying scapular dyskinesis as present or absent promotes better inter-rater agreement when compared to other classifications. This information was added on page 7 (lines 146-147).

- Regarding muscle strength testing, please provide further justification as to why these muscles are tested and not others associated with scapular dyskinesis, such as middle or upper trapezius.

Response: Although the middle and upper trapezius are also scapular stabilizers, the serratus anterior and lower trapezius are the main contributors for scapular upward rotation and posterior tilt, which are the assisted motions during execution of the SAT. We added this information on page 8 (lines 152-154).

- For muscle strength testing, in the case of SAT-positive participants might have evidenced pain in LT strength testing, in what way did they control the appearance of symptomatology?

Response: Unfortunately, appearance of pain during the strength tests was not controlled. In fact, some of the patients reported pain and this may have influenced the strength results. We added this point as limitation of the study on pages 17-18 (lines 366-367).

- Regarding the muscle strength test, the authors state that three repetitions were performed. Please clarify whether the value used in statistical analysis was the average of the records or the maximum value.

Response: We used the average value of the three repetitions in statistical analysis. This information was added on page 8 (lines 172-173).

- Please provide an a priori statistical power analysis in relation to the sample analyzed.

Response: The analysis performed using CART allows the determination of nonlinear relations (interactions) between variables, different from commonly used techniques like linear regression. In addition, it does not require a specified distribution of outcome data or a large sample size, since we have used a leave-one-out cross-validation.

Phelps MC, Merkle EC. Classification and regression trees as alternatives to regression. Proceedings of the 4th Annual GRASP Symposium: Wichita State University, 2008.

Ferreira, Victor MLM, et al. Interaction of foot and hip factors identifies achilles tendinopathy occurrence in recreational runners. Physical Therapy in Sport 45 (2020): 111-119. 

Mendonça, Luciana D, et al. Association of hip and Foot Factors with Patellar Tendinopathy (Jumper's knee) in athletes. Journal of Orthopaedic & Sports Physical Therapy 48.9 (2018): 676-684. 

Haik, Melina N, et al. Biopsychosocial aspects in individuals with acute and chronic rotator cuff related shoulder pain: Classification based on a decision tree analysis. Diagnostics 10.11 (2020): 928

- Please indicate the procedure for selecting CART validation and training data.

Response: The training and validation datasets were selected by using cross-validation, since it is useful and commonly employed in machine learning-based approaches [1]. However, when the number of instances in a data set is small, leave-one-out cross-validation should be used to obtain a reliable accuracy for a CART technique [2]. In leave-one-out cross-validation, the number of folds is equal to the number of instances. This way, the CART was applied once for each instance, using all other instances as a training data, while the selected instance represents the validation data. 

[1] Blockeel, Hendrik, and Jan Struyf. "Efficient algorithms for decision tree cross-validation." Journal of Machine Learning Research 3.Dec (2002): 621-650. 

[2] Wong, Tzu-Tsung. "Performance evaluation of classification algorithms by k-fold and leave-one-out cross validation." Pattern Recognition 48.9 (2015): 2839-2846. 

Results

- Please, with a wide reading audience of the manuscript in mind, improve the presentation of Table 1: the units of the variables should be presented without parentheses (e.g.: Age, years) as it confuses with SD values in subsequent columns. Please remove the acronym SD from the columns, since it is explained in the table caption. Please indicate which p-values correspond to t-test and which to Chi2 test. Submit p-values with 4 significant values.

Response: Alterations were made as suggested.

- The ideas presented in lines 225 to 227 should be included in the discussion section.

Response: This information was included on page 17 (lines 350-353).

Discussion

- The authors propose their discussion by establishing causal relationships between the variables. Given the context of the research, a causal relationship between the clinical variables analyzed is not adequately justified. I suggest toning down a bit the relationships that CART allows to conclude. I suggest reviewing Nogueira et al (2022) Wires Data Mining and Knowledge Discovery DOI: 10.1002/widm.1449, who takes a rather broad look at the problematization of causality and the use of different algorithms to determine it.

Response: Discussion was revised to avoid causal relationships between variables. 

- I suggest that the authors review the limitations of their research in accordance with the comments made.

Response: Done as suggested.

Reviewer #2: Review

What factors contribute to the Scapular Assistance Test result? A classification and regression tree approach

ABSTRACT

1. Describe the group of factors used in the analysis such as biomechanics, psychological ... what else?

Response: The abstract was rewritten.

2. The results about the angular of onset which was the main predictor should be addressed in the abstract.

Response: This information was added in the abstract.

INTRODUCTION

1. The first sentence of second paragraph should be rewritten. The pectoralis minor length is not other factors but it should be considered as the underlying factors influencing the abnormal motion of scapular.

Response: The whole paragraph was rewritten (page 4-5, line 53-80).

2. The underlying impairments reported to contribute to scapular movements and then could influence SAT should be listed. The factors might also should be introduce in categories as the authors addressed in the discussion.

Response: Introduction was rewritten (page 4, line 58-68).

Methods

1. 150 degrees should not be considered as full active elevation

Response: Agree. We deleted “full” (page 5, line 90).

2. Please describe the test procedure of Scapular Assistance Test clearly. Did you have the participants perform movement before assisant was provide. How many times the participants have to perform elevation? When and how you evaluate the pain? Etc

Response: We have improved description of the procedure of SAT on page 6 (lines 119-121 and 124).

3. Did you also assess the pain intensity during SAT. Please report the change of pain intensity in both positive and negative SAT groups.

Response: The test was considered positive when individuals reported a decrease in shoulder pain of two or more points on the 11-point numerical pain rating scale during the assisted elevation as compared to the elevation without assistance. Based on that, pain intensity was asked, but unfortunately not registered. This was stated as a limitation of the study (page 17, lines 363-364).

4. The Scapular dyskinesis test was tested in which plane?

Response: Scapular dyskinesis was tested in the sagittal plane (page 7, line 144).

5. For muscle strength test, Describe the direction of movement that used for producing isometric contraction of SA and LT.

Response: This information was added on page 8 (lines 157, 160-163).

Results

1. Table 1 should present mean±SD and number (%)

Response: Table 1 was changed accordingly.

Discussion

1. What is kgf?

Response: Kilogram force. This information was added in the discussion and table 1.

2. If the scapular dyskinesis is not present, how the assisting force and direction applied during the SAT?

Response: The SAT was always applied towards upward rotation and posterior tilt, independently if scapular dyskinesis was present or absent. 

3. So the subject with high level of pain catastrophizing tend to have negative or positive results of SAT?

Response: Negative. This information is on page 14 (lines 288-301).

4. Please discuss more about clinical implication of the findings. Which factors or outcomes should be considered in people with positive SAT?

Response: This study indicates that SAT result is not explained only by muscle activation/strength impairments or by scapular movement deficits as initially suggested, but from an interaction of different factors including pain catastrophizing, which is a psychosocial aspect. A positive SAT result may result from interaction of scapular dyskinesis, pain catastrophizing, pain at rest and lower trapezius and serratus anterior muscle strength. Interestingly, pain catastrophizing was selected multiple times and may be considered an important predictor for a positive SAT. Therefore, we should consider that previous studies showed that individuals with a positive SAT were more likely to present with reduced scapular posterior tilt (Ribeiro et al., 2020) and shortening of the pectoralis minor muscle (Grimes et al., 2020). This information is on page 16 (lines 320-325)

Overall

Please recheck the grammar and format of inserted references.

Response: Done.

Reviewer #3: Review Comments to the Author

Thank you for the opportunity to review the manuscript. The study aims to investigate determine predictive factors related to the SAT in individuals with shoulder pain during arm elevation, and to analyze how these predictors interact in a nonlinear manner to discriminate the result of a positive and negative SAT. These factors are explored with a decision tree analysis.

The manuscript is well written and described in adequate details for the most part. Please find below my comments for feedback.

Response: Thank you for your time and effort in reviewing this manuscript. We expect to attend your concerns.

Methods&Results

・This study`s include criteria was “Individuals were included if they had atraumatic self-reported unilateral shoulder pain during flexion of the arm for at least four weeks since first onset and full active arm flexion (~150°) as measured by a clinical inclinometer”. You should add an explanation as to why this was the inclusion criteria. (Example. Non-traumatic or traumatic, duration of onset: 4weeks, etc.)

Response: This study is part of a larger project conducted by the same research group. This information was added on page 5 (lines 92-93). Scapular kinematics was tracked in the larger project where active arm flexion (~150°) was needed for a complete set of kinematic data. Non-traumatic shoulder pain is highly common in the clinical practice, and 4 weeks of duration was determined to guarantee the pain was not something occasional.

・This study described the limitation section. “Although a 2-point decrease in pain intensity was considered for a positive SAT, a register before and after the SAT was not registered”. The NRS score at baseline is likely to affect the SAT result. Is the NRS score not included in the inclusion criteria for this study? If it is not included in the uptake criteria, I think it should be added to the limitation section.

Response: The test was considered positive when individuals reported a decrease in shoulder pain of two or more points on the 11-point numerical pain rating scale during the assisted elevation as compared to the elevation without assistance. Based on that, pain intensity was asked, but unfortunately not registered. This was stated as a limitation of the study (page 17, lines 363-364).

・In Results, the predictive model obtained an average accuracy of 92.9％ and ROC area of 0.96. This model indicated good accuracy, but split was complexity. Therefore, it is difficult to interpret this model. How were splits selected in the tree (entropy or complexity penalty)? Also, the splits seem a little nuanced, given that the same variable is used more than once. Can the authors write a little about whether reported splits would be clinically relevant or significant?

Response: We respectfully disagree about the model complexity, since it used only 6 (six) variables in the splits. However, it has 7 (seven) levels of depth, which really causes, at first, difficulty in reading/understanding. Thus, it is important to make it clear that a set of 5 to 7 questions would be enough for the physical therapist to obtain a clinical answer that would guarantee the individual's SAT classification, needing to obtain only the 6 variables mentioned above. Additionally, it can be highlighted that, in this division, there is not a p value associated with. However, the nodes are hierarchically ordered, being the angular onset of pain the most important predictor. Relating to the split technique, it is important to clarify that decision trees commonly employ Information Gain (entropy concept) to perform the splits, while the complexity penalty is generally used as pruning algorithm. In this sense, we can say that both techniques were used.

Discussion

・I believe that SAT positivity is a test to indicate scapular dysfunction, but the results of this study included SAT-positive participants with high muscle strength. The author may write some thoughts on how SAT positivity should be interpreted and utilized in clinical practice, including muscle strength?

Response: The SAT was first described to assess if scapular dysfunction may be associated with shoulder pain. According to Kibler (1998), a positive SAT may suggest poor strength and/or activation of the serratus anterior and lower trapezius muscles, which are important scapula movers. Although the participants in this study presented high muscle strength, we cannot talk about muscle activation. Serratus anterior and lower trapezius muscles may not be properly activated during elevation of the arm. Our findings show that a positive SAT may result from interaction of scapular dyskinesis, pain catastrophizing, pain at rest and lower trapezius and serratus anterior muscle strength. Interestingly, pain catastrophizing was selected multiple times in the tree and may be considered an important predictor for a positive SAT. Clinicians should also be aware that previous studies have shown that individuals with a positive SAT are more likely to present greater scapular anterior tilt (Ribeiro et al., 2020) and pectoralis minor tightness (Grimes et al., 2021). This information was added on page 16 (lines 320-325).

・Looking at the PCS results (e.g., SAT negative with PCS score ≤ 8 points and > 35 points), I do not think the PCS score determines SAT positives and negatives. Please check my comments about decision tree complexity. If the PCS score does affect the SAT results, can the author speculate on the mechanism why this might be the case? Since the PCS score is also lower than the cutoff, it is by no means clear that a high PCS score affects the SAT results. PCS score is also lower than the cutoff value, so we cannot say that the high PCS score affects.

Response: The negative SAT group was more catastrophizer than the positive SAT group when the descriptive characteristics of both groups were compared. However, in the CART, the cut-off values of the PCS depend on the interaction with the other variables (angular onset of pain, scapular dyskinesis, pain at rest and lower trapezius and serratus anterior muscle strength). Individuals with negative SAT frequently presented higher score in the PCS scale in the CART (in the nodes). One thing that is important to note is that the PCS scale is divided into three domains: hopelessness, magnification and rumination. However, we just considered the total score, but individuals with negative SAT may present higher levels of hopelessness and assisting the scapula may not to modify the pain of these individuals with negative SAT. Future studies should investigate the influence of each domain in individuals with shoulder pain.

Reviewer #4: Thank you to the authors for their submission. The paper addresses an important clinical and research topic and I am happy to review this paper again after the authors have addressed my concerns.

Response: Thank you for your time and effort in reviewing this manuscript. We expect to attend your concerns.

I suggest to include some sample characterization in the abstract (at least age). In the first paragraph of introduction, I suggest to bring in a more detailed way what is the “problem in science” (relevance).

Response: Sample characterization was added in the abstract. The introduction was changed to bring the relevance of the study. We hope it is clear now.

Authors should consider to add more information from references 10 and 11 about Other variables as serratus anterior and lower trapezius muscle strenght in the introduction. It should be clear in the introduction whats the importance (what Fields knows and remining gap?) of adding these variables to the model. 

Response: The introduction was rewritten.

Only the pectoralis minor lenght was mentioned and this variable has not been evaluated.

Response: The introduction was rewritten.

The authors mentioned that all data were presetend in the manuscript. However, in addition to summary statistics, the data points behind means and standard deviation also should be avaiable as a requirement from the Journal.

Response: We will release all data maintaining the anonymity of participants if that is required. We will follow the requirement from the Journal.

I would suggest to add more information in the results description.

Response: We would be glad to add, but we could not figure out what the reviewer is missing.

Lines 243-245: I would suggest to make this sentence/association clearer and also add the reference.

Response: This part of the text was rewritten.

Line 248 – “the pain are likely dye to altered scapular motion” – I suggest to rewrite this sentence considering the several factos involved in the chronicity of atraumatic shoulder pain (patients had an average of 27 months of pain duration).

Response: This sentence was rewritten.

Lines 249 - 252: please, consider to make clearer the findinds interpretation: increased pain catastrophising, increased ? pain at rest, decreased serratus anterior strenght.

Response: This sentence was rewritten.

Lines 263 – 266: The study design and the findings presented may not support this suggetion.

Response: We agree and have deleted this sentence.

Line 267: “mediating effect”: I suggest a clearer definition of this term.

Response: This sentence was rewritten.

---

## [Editor Report · Decision Letter 1]

12 Oct 2022

What factors contribute to the Scapular Assistance Test result? A classification and regression tree approach

PONE-D-21-38851R1

Dear Dr. Larissa Pechincha Ribeiro,

We’re pleased to inform you that your manuscript has been judged scientifically suitable for publication and will be formally accepted for publication once it meets all outstanding technical requirements.

Kind regards,

Juan Guerrero-Henriquez

Guest Editor

PLOS ONE

Additional Editor Comments (optional):

Dear Larissa,

Thank you for your submission to PlosOne.

I am writing to inform you that your manuscript - What factors contribute to the Scapular Assistance Test result? A classification and regression tree approach - has been Accepted for publication. Congratulations!

---

## [Editor Report · Acceptance letter]

14 Oct 2022

PONE-D-21-38851R1 

What factors contribute to the Scapular Assistance Test result? A classification and regression tree approach 

Dear Dr. Pechincha Ribeiro:

I'm pleased to inform you that your manuscript has been deemed suitable for publication in PLOS ONE. Congratulations! Your manuscript is now with our production department. 

Kind regards, 

on behalf of

Dr. Juan Guerrero-Henriquez 

Guest Editor

PLOS ONE